# Process Maps for Predicting Austenite Fraction (vol.%) in Medium-Mn Third-Generation Advanced High-Strength Steels

**DOI:** 10.3390/ma17050993

**Published:** 2024-02-21

**Authors:** Azin Mehrabi, Hatem S. Zurob, Joseph R. McDermid

**Affiliations:** Department of Materials Science and Engineering, McMaster University, 1280 Main Street West, Hamilton, ON L8S 4L8, Canada; mehraba@mcmaster.ca (A.M.); mcdermid@mcmaster.ca (J.R.M.)

**Keywords:** processing map, medium-Mn third-generation advanced high-strength steels, intercritical austenite vol.%, retained austenite vol.%, DICTRA

## Abstract

Process maps were developed using a combination of microstructural analysis and DICTRA-based modeling to predict the austenite vol.% as a function of the intercritical annealing parameters and starting microstructure. The maps revealed a strong dependence of the calculated austenite fraction (vol.%) on the Mn content (4–12 wt.%) and intercritical annealing temperatures (600 °C to 740 °C). The calculations were carried out for constant carbon, Al, and Si contents of 0.2 wt.%, 1.5 wt.%, and 1.0 wt.%, respectively. A modified empirical equation proposed by Koistinen and Marburger was employed to calculate the room-temperature retained austenite vol.% as a function of the intercritical annealing temperature, including the effect of the austenite composition. The process maps offer valuable insights for designing intercritical treatments of medium-Mn steels, aiding in the optimization of steel properties for automotive applications.

## 1. Introduction

There is an increasing demand for high-strength steels in the automotive industry in order to enable weight reduction and improved collision resilience [1]. This has led to the development of advanced high-strength steels (AHSS) with excellent strength and formability characteristics. In recent decades, considerable efforts have been dedicated to enhancing the mechanical properties of these steels by optimizing the composition and thermomechanical processing [2,3]. Austenitic, second-generation steels with Mn contents in the range of 17 to 30%, received a lot of attention in the 1990s and 2000s due to their superior balance of strength and elongation. However, their engineering applications have been limited owing to high alloying costs and production problems [4].

More recently, much attention has been paid to AHSS with medium Mn (med-Mn) addition [5,6,7]. Med-Mn steels generally contain 0.05–0.4 wt.% C, 4–12 wt.% Mn, 0.5–3 wt.% Al, 0.5–2 wt.% Si, and small amounts of micro-alloying elements, such as Mo, Ti, V, and Nb [8,9,10]. Med-Mn steels typically have a complex microstructure consisting of ferrite and/or martensite, and 20–50 vol.% retained austenite (RA). The presence of retained austenite is important for enhancing the mechanical properties of med-Mn steel due to the transformation-induced-plasticity (TRIP) effect. In these steels, the higher concentrations of Mn result in retaining a considerable fraction of intercritical austenite and by consequence a desirable strength–ductility balance [11,12,13,14]. One of the first reported results on the effect of increasing alloy Mn concentration is attributed to Miller in the 1970s [15]. It was reported that a high UTS of 1150 MPa with a uniform elongation over 25% could be achieved for a cold rolled 0.1C-6Mn alloy after annealing at 640 °C for 1 h. This was attributed to the high fractions of metastable RA which transformed to martensite during deformation. 

In the processing of med-Mn TRIP steels, the formation of austenite primarily occurs through intercritical annealing (IA). During IA, the controlled partitioning of C and Mn into the austenite increases its chemical stability, leading to the partial retention of austenite at room temperature. The volume fraction of intercritical austenite increases with increasing annealing temperature. Increasing the intercritical temperature, however, decreases the average C and Mn contents of the intercritical austenite and, as a result, reduces its chemical stability. As a result, significant fractions of the austenite formed at higher intercritical temperatures transform to martensite during cooling [16,17,18]. The optimum annealing temperature is determined by achieving the desired balance between the volume fraction and the stability of the austenite [19]. Designing precise alloying and processing strategies is critical to control the volume fraction of RA and, consequently, the mechanical properties of the steel. 

It is important to note that steels destined for automotive applications must be corrosion-resistant as they can be exposed to aggressive environments—e.g., roads with heavy loads of de-icing salts. Continuous galvanizing lines (CGLs) play an important role in providing this corrosion protection in automotive steel manufacturing due to their high production capacity and cost-effectiveness. It is, therefore, necessary to integrate med-Mn steels into the automotive industry with thermal processing parameters compatible with industrial CGL processing windows. The continuous galvanizing process is described in detail in [20]. The steel undergoes a short thermal treatment, typically lasting 3–5 min depending on the line speed, in the radiant tube heating and soaking sections of the CGL. To meet the CGL’s productivity targets, the annealing parameters must fall within this time range while achieving the desired microstructures and mechanical properties. Although several studies [21,22,23,24,25] have found that med-Mn steels can achieve 3G AHSS properties through intercritical annealing, the long thermal treatments employed in these studies are not compatible with the industrial CGL. For the present experimental alloy, significant volume fractions of chemically stable retained austenite were obtained after annealing at 710 °C for 120 s, which corresponds to continuous galvanizing industrial practices [26].

In the present study, we focus on developing process maps for the formation of austenite during IA. The maps were constructed using a combination of microstructural analysis and DICTRA-based modeling for different starting microstructures. The process maps are essential for determining the optimal IA parameters for med-Mn steels with specific chemical compositions. This knowledge is crucial for further development of the steel properties to meet the requirements of automotive applications. Also, a modified calculation method [27] is used to predict the room temperature retained austenite fraction as a function of the IA temperature. Additionally, the optimal annealing temperature range to achieve the desired balance between the volume fraction and the stability of the retained austenite in med-Mn steels has been suggested. 

## 2. Overview of Models

### 2.1. DICTRA-Based Model

The DICTRA module of Thermo-Calc was used to model the austenite growth kinetics for both the as-cold-rolled condition with a tempered martensite (TM) microstructure and a fully austenitized and as-quenched martensite (M) starting microstructure, which is known to have more rapid austenite reversion kinetics [26,28]. DICTRA is an engineering tool for diffusion-based simulations in multicomponent alloys [29,30]. The software uses the Thermo-Calc thermodynamic (TCFE12) and mobility (MOBFE5) databases [29], which have been derived from assessed experimental data in the literature. The simulations are one-dimensional. DICTRA can accommodate planar, cylindrical, and spherical geometries, all of which can be reduced to a single space variable. In order to set up the simulations of austenite growth, the morphology, compositions, and size of phases are required. The diffusion of C, Mn, Si, and Al was considered in the simulation. The size of the simulation cell was based on the observed length scales within the TEM micrographs reported previously [26,28,31]. 

Two initial configurations were considered for the DICTRA simulation. In the case of the TM starting microstructure, austenite (γ) nucleated at ferrite (α) grain boundaries, after cementite dissolution had taken place (Figure 1a) [31]. A total of 50 and 150 grid nodes were distributed in austenite and ferrite, respectively. In the case of the M starting microstructure, austenite grew from the existing inter-lath RA in martensite (Figure 1b) [31]. The selection of the simulation cell was guided by the half-width of a representative martensite lath from the initial microstructure. A total of 150 and 500 grid nodes were distributed in austenite and ferrite, respectively. Based on the experimental results, the concentration of C in austenite was determined to be 1.5 wt.% [32]. The Al and Si contents in austenite were assumed to align with the nominal alloy composition. The composition of ferrite was then determined based on the mass balance.

Martensite is not categorized as a separate phase in DICTRA. Thus, in common with others in the literature [33,34,35,36,37], the BCC phase with a high carbon content is utilized to represent martensite in the model. Throughout this study, both phases will be identified as α.

### 2.2. Koistinen–Marburger Model

The Koistinen and Marburger empirical formula [38] was used for calculating the volume fraction of RA:(1)$$f_{y}=exp\;[-m(M_{s}-T{)}^{n}$$
where ***f_y_*** is the volume fraction of RA and ***T*** denotes the lowest temperature reached during quenching (in the present case, room temperature). ***m*** and ***n*** are coefficients linked to transformation kinetics. ***M_s_*** is the martensite start temperature. The parameters ***m*** and ***n*** were adjusted as a function of the Mn and C content of the austenite in wt.% [39].
(2)m=0.0076+0.0182C−0.00014Mn
(3)n=1.4609+0.4483C−0.0545Mn

The martensite start (***M_s_***) temperature is determined by applying the following equation [39,40]: (4)$$M_{s}\left({}^{\circ}C\right)=545-423C-30.4Mn-7.5Si+30Al-60.5V_{\gamma }^{-1/3}$$
where ***V_γ_*** is the average grain volume of austenite in µm^3^. The Mn, C, Si, and Al contents in wt.% were identified with those present on the austenite side of the interface after holding for 120 s at the intercritical annealing temperature.

Based on the experimental observations conducted on med-Mn steels [31], a fixed austenite grain size of 1 µm was considered for the simulations.

The composition of the intercritical austenite at 120 s was calculated using the DICTRA module of Thermo-Calc. This composition was used to determine the values of ***m***, ***n***, and the ***M_s_*** temperature. The DICTRA simulations also provided the volume fraction of austenite as a function of IAT and holding time. The room temperature austenite fraction after quenching is expressed using
(5)$$f_{ret\gamma }=f_{\gamma }\times f_{D\gamma }$$
where ***f_ret γ_*** is the retained austenite volume fraction at room temperature, ***f_γ_*** is the volume fraction of austenite after quenching, as calculated by Equation (1), and ***f_Dγ_*** is the austenite volume fraction at the IA temperature, as calculated by DICTRA. 

The Mn, C, Si, and Al contents were determined using the DICTRA software version 2023.2.119013-24 and the MOB-FE5 mobility database. The composition of the austenite after annealing for 120 s was estimated. The diffusion of Mn, C, and Al resulted in the formation of a Mn and C-enriched region and an Al-depleted region on the austenite side of the interface. DICTRA predicted very limited partitioning of Si between the austenite and ferrite. The carbon and Mn content in the intercritical austenite increased to ranges of 0.3–0.5 wt.% and 7–11 wt.% in both starting microstructures, respectively, due to solute partitioning during the transformation. The calculation results are in good agreement with measured values reported in [26].The Al content near the interface of the intercritical austenite also decreased to a range of 1–1.4 wt.%. 

## 3. Results

A set of process maps were developed for a prototype med-Mn steel with the generic composition of 0.2C-*x*Mn-1.0Si-1.5Al wt.% (*x* = 4–12 wt.%). These maps predict the vol.% of austenite formed during IA, based on the intercritical annealing temperature (IAT), IA time, and the initial microstructure.

The process maps were developed for an IAT range of 600–740 °C, while the Mn content was varied between 4 and 12 wt.%. Notably, the process map specifically demonstrated the austenite vol.% resulting from an IA holding time of 120 s, which is compatible with industrial continuous galvanizing practices and had been shown to produce significant volume fractions of chemically stable RA and attractive 3G properties [26,28,41]. It should be recalled that the vol.% of RA at room temperature was calculated as a function of IAT and Mn content using Equations (1)–(5).

### 3.1. Tempered Martensite Starting Microstructure

#### 3.1.1. Intercritical Austenite vol.% Calculation Using DICTRA

The TM starting microstructure model simulates austenite growth in a globular microstructure, as shown in Figure 1a. Based on the equiaxed microstructure, a spherical geometry was considered [42]. Figure 2 depicts an isopleth of the Fe-0.2C-*x*Mn-1.0Si-1.50Al (wt.%, where *x* = 0–12 wt.%) phase diagram showing the temperature range over which cementite is dissolved. The TM starting microstructure simulations were conducted in the temperature and Mn concentration range in which cementite dissolution had been completed. For this reason, IA temperatures greater than 680 °C were used. The total cell size changed from 330 nm to 440 nm in the TM starting microstructure as the IA temperatures increased from 680 °C to 710–720–740 °C, respectively. This change was introduced in an attempt to capture the coarsening of the microstructure with increasing temperatures and was guided by earlier experimental observations [26,28,31]. 

Figure 3b shows the intercritical austenite vol.% calculated for the TM starting microstructure as a function of the Mn content of the alloy and IAT for an IA time of 120 s. Throughout the analysis, the bulk C, Al, and Si contents were held constant at 0.2 wt.%, 1.5 wt.%, and 1.0 wt.%, respectively, while the Mn content increased from 4 to 12 wt.%. As expected, an increasing trend in austenite vol.% was observed with increasing IAT from 680 °C to 740 °C. For IATs of 720 °C and 740 °C, for alloy Mn contents of 10 wt.% and 9 wt.%, respectively, the amount of austenite reached nearly 100% after a 120 s holding period. 

As a reference point, the equilibrium vol.% of austenite as a function of IAT predicted using Thermo-Calc is shown in Figure 3a. The calculated vol.% of austenite after an IA of 120 s (and prior to quenching) is shown in Figure 3b. A comparison of the two maps shows that there is a significant difference between the equilibrium values and the modelled values for an IA of 120 s. The difference will likely be even greater for shorter holding times. Differences are smaller at higher IATs, while they are greater at lower IATs. The calculated vol.% of intercritical austenite after 120 s is higher than the predicted equilibrium values at 720 °C and 740 °C. This highlights that equilibrium calculations cannot be used to calculate the amount of austenite formed during intercritical annealing. 

In situ high-energy XRD measurements were conducted to compare the intercritical austenite vol.% during a 120 s IA at 710 °C for a 0.15C–5.56Mn–1.89Al–1.1Si steel. The results in Table 1 show that the measured intercritical austenite vol.% was approximately 40% at 710 °C, which is in good agreement with the calculated data.

#### 3.1.2. Retained Austenite Calculation

In Figure 3c, the calculated room temperature retained austenite vol.% is plotted for intercritical annealing temperatures of 680, 700, 720, and 740 °C. It is worth noting that the maximum value of the room temperature retained austenite vol.% is achieved at an IAT temperature of 680 °C. At 680 °C, the vol.% of RA closely corresponds to that of intercritical austenite within the Mn content range of 5 to 6.5 wt.% (Figure 3b,c).

### 3.2. Martensite Starting Microstructure

#### 3.2.1. Intercritical Austenite vol.% Calculation Using DICTRA

The M starting microstructure model simulates austenite growth from the lamellar geometry illustrated in Figure 1b. A planar geometry originates from the inter-lath RA geometry observed in previous work on this family of alloys [26,28,31,41]. The cell size for the DICTRA simulations for the 600–660 °C and 680–740 °C IATs were approximated as 200 nm and 300 nm in the M starting microstructure, respectively. The larger cell size at the higher IAT temperatures reflects the coarsening of the microstructure with higher IA temperatures. The simulation cell sizes were chosen based on the microstructural observations that have previously been made [26,28,31]. Figure 4a shows the calculated intercritical austenite vol.% as a function of IAT and Mn content. The simulation predicts 12 to 20 vol.% intercritical austenite at an IAT of 600 °C with increasing alloy Mn, and 55 to approximately 100% at 740 °C. 

The austenite vol.% measured using in situ high-energy XRD during IA at 665 °C and 710 °C for 120 s for the 0.15C–5.56Mn–1.89Al–1.1Si steel are shown in Table 1 for comparison. The results show that the measured intercritical austenite vol.% was roughly 31% ± 2 at 665 °C and 45% ± 2 at 710 °C, which is in good agreement with the simulation results [32].

#### 3.2.2. Retained Austenite Calculation

Figure 4b illustrates the predicted RA vol.% obtained using Equation (1) for the Fe-0.2C-*x*Mn-1.5Al-1Si wt.% steels (*x* = 4–12 wt.%) with a martensitic starting microstructure from 680 °C to 740 °C. The effect of alloying on the austenite transformation kinetics is also clearly expressed in the values for m and n (Equations (1) and (2)). With increasing Mn content, the kinetics of the austinite transformation are accelerated as Mn is a strong austenite stabilizer. [11]. At 680 °C, the intercritically annealed austenite is fully retained after quenching to room temperature, provided that the alloy Mn content is between 5 and 7.5 wt.% (Figure 4).

### 3.3. Comparing the Model with the Literature

The process maps for intercritical austenite and RA provide the overall trends for the evolution of austenite vol.% as a function of steel composition and intercritial temperature. There is limited research that matches the exact conditions used to create the maps (for example, some data will match the composition, but not the temperature or the holding time). In order to validate the model, we carried out simulations that exactly match the experimental compositions and intercritical times/temperatures in the available literature [26,28,31]. Table 1 presents a comparison between the measured values of RA vol.% obtained from data in the literature [26,28,31] and the corresponding calculated values derived from both the DICTRA-based model and the Koistinen–Marburger model. Table 1 shows the experimentally measured RA austenite after annealing at 665, 675, 690, and 710 °C for two different med-Mn steels [26,28]. The RA calculated using Equation (1) is observed to be in excellent agreement, differing by only ±4% when compared to the experimental results. It is worth noting that in the calculation of RA using Equation (1), Cr is also taken into account for ***M_s_*** determination for the 0.18 C–5.91 Mn–1.5 Si–0.4 Al-0.6Cr steel.

## 4. Discussion

A comparison between the calculated intercritical austenite values from DICTRA (Figure 3b and Figure 4a) and the equilibrium austenite values (Figure 3a) reveals that equilibrium (thermodynamic) calculations cannot be used to estimate the vol.% of austenite that forms during intercritical annealing under practical processing conditions. Thus, a comprehensive calculation is indispensable. The DICTRA-based model combined with the Koistinen–Marburger model was utilized to better estimate the austenite vol.% in med-Mn steels with as-quenched martensite and tempered martensite starting microstructures. 

The calculations based on DICTRA provide a good prediction of the intercritical vol.% of austenite. In order to validate the maps during IA, high-energy X-ray diffraction (HEXRD) tests were conducted and are reported in Table 1 for the 665 °C × 120 s and 710 °C × 120 s IA M samples and the 710 °C × 120 s IA TM sample in 0.15C–5.56Mn–1.89Al–1.1Si steel [32]. From this, it can be seen that the experimental intercritical austenite vol.% for the 0.15C–5.56Mn–2Al–1Si steel is in good agreement with the DICTRA-predicted intercritical austenite vol.%. The effect of the starting microstructure on austenite transformation kinetics during IA can be concluded by comparing the two process maps in Figure 3b and Figure 4a. During IA for the M starting microstructure, a significantly larger vol.% of intercritical austenite is formed versus the same IAT combination for the TM starting microstructure. This is due to the presence of a lath-shaped martensite starting microstructure. The Mn diffusivity is much higher in the martensite microstructure, resulting in a higher concentration of Mn in the austenite [43]. 

Furthermore, the DICTRA-based process maps for intercritical austenite indicate that slight changes in temperature can lead to notable changes in the vol.% of intercritical austenite for both starting microstructures. This finding emphasizes the importance of carefully controlling temperature during the IA process to achieve the desired microstructural characteristics. Although these maps are valuable tools for designing intercritical treatments, they do not provide a complete answer regarding the final vol.% of chemically and mechanically stable RA at room temperature. This is because the quantity and stability of RA are influenced by various factors, including the chemical composition of the austenite (e.g., C/Mn content) [37,44]. The calculated results reflect the vol.% of austenite before quenching to room temperature. However, after quenching, a portion of the intercritical austenite is expected to transform into martensite. In order to address this limitation, the RA maps have been computed using the Koistinen–Marburger model for the med-Mn steels as a function of the starting microstructure.

The combination of the Koistinen–Marburger model, DICTRA, and X-ray diffractometry (XRD) was employed to determine the amount of retained austenite. It should be mentioned that to calculate the RA using Equations (1)–(5), the average grain volume of austenite is needed. Based on the experimental observations conducted on med-Mn steels [31], a constant grain size of 1 µm at all IATs was considered for the austenite. Lee et al. [27] studied the effect of varying grain sizes at annealing temperatures on the calculated ***M_s_*** temperature. Their findings indicate that, at high annealing temperatures, grain size does not change the calculated ***M_s_*** temperature, and the stability of austenite is no longer controlled by its grain size. For validation, values of experimentally measured RA vol.% via XRD for various prototype med-Mn steels from the literature [26,28,31] are compiled in Table 1. The validation data were selected to limit the study to two variables: temperature and Mn content. This required fixing the annealing times to be compatible with CGL thermal treatments and compositions similar to those of the experimental alloy with respect to C, Al, and Si contents. The literature data selected were chosen based on their compatibility with the fixed annealing time and compositional constraints documented in Table 1 [26,28,31]. Several studies on austenite growth in med-Mn steels, compatible with industrial CGL thermal treatments, have been conducted [7,45,46,47], while the intercritical annealing time differed from the model. As shown in Table 1, the model values are in good agreement with the experimentally measured vol.% of RA obtained from the literature. The variation in grain size between the experimental results and the calculated RA could be the primary reason for the slight difference. The results reveal that the process maps provide valuable insights into designing intercritical treatments. These maps cover a range of med-Mn steel compositions, including 4–12 wt.% Mn, 1.5 ± 0.5 wt.% Al, 1 ± 0.5 wt.% Si, and 0.15–0.2 wt.% C. Changing the alloy C, Al, and Si content significantly affects both the thermodynamic and kinetic calculations [13]. It should also be mentioned that some caution should be exercised using the process maps for higher Mn grades as the starting microstructures may be duplex in nature. Thus, further development of the process simulations to take into account the chemical stability of the austenite and for higher Mn alloys will be undertaken.

When examining Figure 3c and Figure 4b, which illustrate the relationship between RA vol.%, IAT, and alloy Mn content, it becomes evident that there is much more RA in the M starting microstructure compared to the TM starting microstructure for a given 120 s IAT. This is attributed to two reasons. The first is the greater amount of austenite which is formed after 120 s of intercritical annealing in the M starting microstructure compared to the TM microstructure. Secondly, the austenite formed during the intercritical annealing of the as-quenched martensite is more stable than that formed during the intercritical annealing of the tempered martensite. This leads to more retained austenite at room temperature for the steels with the as-quenched martensitic microstructure compared to those with the tempered martensite structure. This finding is consistent with previous research papers [21,28,31,48], and can be linked to C and Mn partitioning from the supersaturated martensitic matrix and the reduction in strain energy connected to the formation of inter-lath austenite films [49]. The presence of inter-lath RA films along the martensite lath boundaries accelerates the start of the reverse transformation by eliminating the incubation time associated with the nucleation step [50]. Upon comparing Figure 3c and Figure 4b, it becomes evident that the RA formed from the M starting microstructure exhibits greater stability compared to that formed from the TM starting microstructure. This increased stability can be attributed to the greater C and Mn enrichment in the RA, a result of the shorter diffusion path in the lath-shaped microstructure compared to the equiaxed microstructure [26,28]. Consequently, this leads to a higher vol.% of RA in the samples with a martensitic starting microstructure. 

By contrast, the formation of austenite in the TM starting microstructure is controlled by the kinetics of cementite dissolution and the availability of carbon to enrich the austenite. The IATs applied to the TM starting microstructure correspond to the range where cementite dissolution occurs. In the TM starting microstructure, a nucleation step and greater diffusion distance compared to the M starting microstructure led to a lower vol.% of chemically stable RA. As the annealing temperature rises, the volume of austenite increases. However, when we elevate the intercritical temperature, it leads to a decrease in the average C and Mn content within the austenite, causing a reduction in its chemical stability. As a result, a significant portion of austenite formed at high IAT undergoes a transformation into martensite during the cooling process [16,17,18]. At the lower IA temperatures, the C content of the intercritical austenite will be higher, resulting in retained austenite at room temperature that aligns reasonably well with the model. The optimal annealing temperature range to achieve the desired balance between the vol.% and the stability of austenite is determined to be 700 °C for both starting microstructures. It is noticeable that the map remains relatively flat at lower Mn contents, suggesting that varying Mn content between 5 to 7% does not notably impact the amount of RA for either starting microstructure. Therefore, designing med-Mn steels with lower Mn contents, around 5 wt.%, can reduce cost while maintaining the same amount of RA as 7% Mn steel. The main challenge is precise temperature control during the thermal processing of med-Mn steels, as even a 20 °C change significantly affects the amount of RA.

It is noticeable that the predicted amount of intercritical austenite (Figure 3b and Figure 4a) increased above that predicted at equilibrium (Figure 3a) at higher IAT. The formation of intercritical austenite beyond the equilibrium amount is attributed to the significantly slower substitutional diffusion within the growing austenite compared to ferrite. In the final stage of austenite growth, its kinetics are controlled by the diffusion of substitutional elements in austenite for final equilibration, potentially leading to austenite shrinkage [50,51]. This shrinkage becomes more pronounced at higher temperatures and higher Mn content. The diffusivities of alloying elements in the parent and growing phases, the composition of the alloy, and IAT can change the amount of excess austenite [50,51,52,53]. It is worth noting that in industrial processes, both the IAT and time are restricted, which means that the equilibrium phase fractions, and the equilibrium chemistry cannot be achieved.

## 5. Conclusions

The development of process maps through a combination of microstructural analysis and DICTRA-based modeling yielded valuable insights into the behavior of med-Mn steels during intercritical annealing. These maps unveiled significant dependencies on Mn content and intercritical annealing temperatures, providing crucial information for the further development of the steel properties in automotive applications. The utilization of the Koistinen–Marburger model deepened our understanding of RA formation kinetics in different starting microstructures. 

Compared to the TM starting microstructure, the M starting microstructure promotes a higher vol.% of intercritical austenite during annealing. Additionally, it leads to a higher RA vol.% after quenching to room temperature. These findings highlight the importance of the M starting microstructure in influencing the microstructural evolution and mechanical properties of the med-Mn steels. 

Overall, these findings contribute to a model-based approach to develop a robust process window for the further development of med-Mn steels, aiding in the design of steel properties tailored for automotive applications. 

## Figures and Tables

**Figure 1 materials-17-00993-f001:**
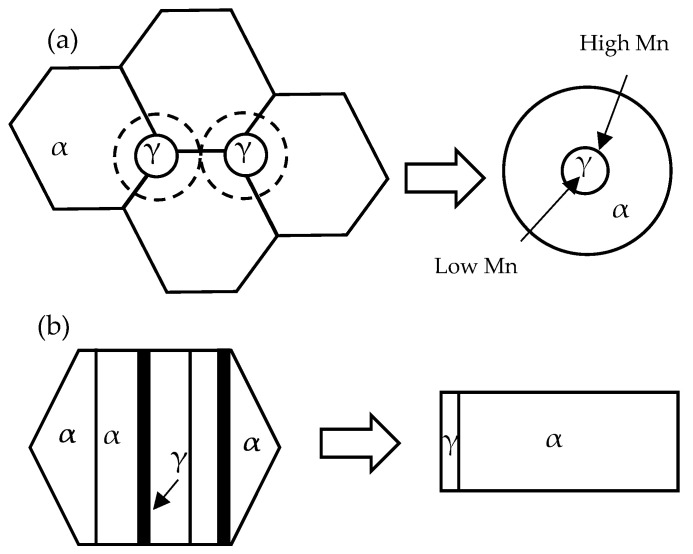
Schematic of initial microstructural configuration and resultant DICTRA geometry for austenite formation during heating: (**a**) TM starting microstructure—austenite nucleation on ferrite with dissolved cementite; and (**b**) M starting microstructure—austenite growth on ferrite (martensite) lath boundaries (γ = austenite, α = ferrite). The arrows indicate regions containing high and low Mn contents. Dashed lines surrounding the γ phase represent the growth of austenite.

**Figure 2 materials-17-00993-f002:**
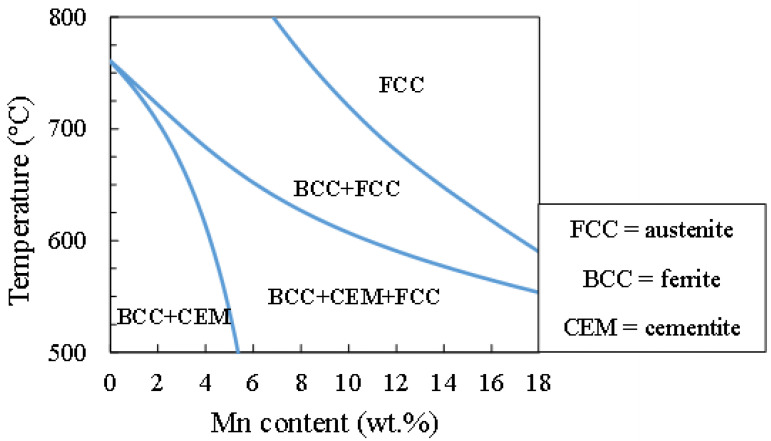
Section of the Fe-0.2C-*x*Mn-1.0Si-1.50Al (wt.%) phase diagram, where Mn is varied from 0 to 12 wt.%. Diagram generated using the Thermo-Calc TCFE12 database.

**Figure 3 materials-17-00993-f003:**
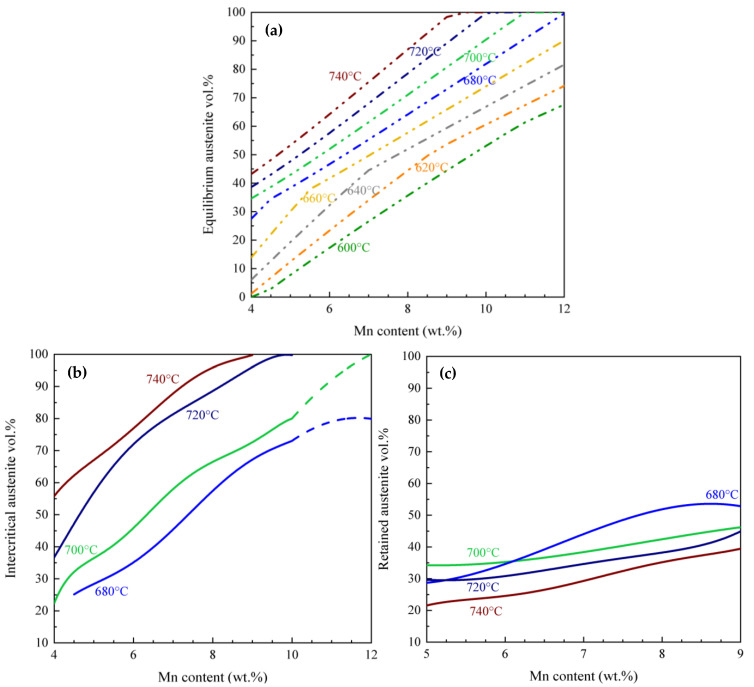
(**a**) Equilibrium austenite vol.%, (**b**) intercritical austenite, and (**c**) calculated RA fraction (vol.%) for Fe-0.2C-*x*Mn-1.5Al-1Si wt.% steels (*x* = 4–12 wt.%) as a function of IAT from 680 to 740 °C for the TM starting microstructure, and assuming an austenite grain size of 1 µm. Note that the dotted lines are related to higher Mn contents as the starting microstructures may be duplex.

**Figure 4 materials-17-00993-f004:**
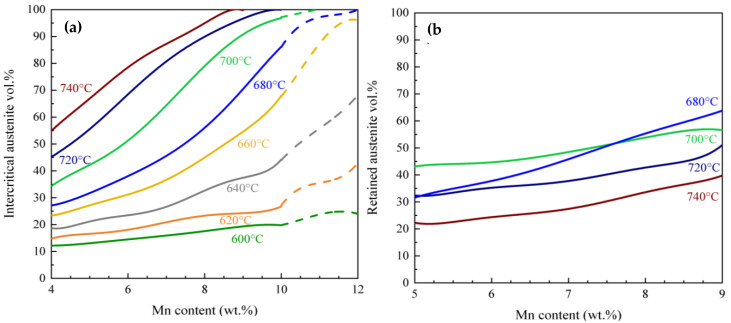
Effects of chemical composition on (**a**) the intercritical austenite and (**b**) calculated RA fraction (vol.%) for Fe-0.2C-*x*Mn-1.5Al-1Si wt.% steels (*x* = 4–12 wt.%) as a function of IATs from 600 to 740 °C for the M starting microstructure, and assuming an austenite grain size of 1 µm. Note that the dotted lines are related to higher Mn contents as the starting microstructures may be duplex.

**Table 1 materials-17-00993-t001:** Comparison of experimental measurements with calculated values for intercritical annealing austenite vol.% versus the final vol.% of retained austenite for several med-Mn steels. Note that all intercritical annealing treatments are for 120 s at the stated IAT.

Composition	Starting Microstructure	IAT (°C)	Measured γ (vol.%)	Calculated γ (vol.%)	Measured RA (vol.%)	Calculated RA (vol.%)
0.18 C–5.91 Mn–1.5 Si–0.4 Al-0.6Cr [28]	TM	675			15.0	19.6
0.15C–5.56Mn–1.89Al–1.1Si [31]	TM	710	40.0	41.8	27.0	30.5
0.18 C–5.91 Mn–1.5 Si–0.4 Al-0.6Cr [28]	TM	710			25.0	23.3
0.15C–5.56Mn–1.89Al–1.1Si [26,32]	M	665	30.8	25.0	21.0	22.3
0.15C–5.56Mn–1.89Al–1.1Si [26,32]	M	710	45.0	42.0	31.0	32.5
0.18 C–5.91 Mn–1.5 Si–0.4 Al-0.6Cr [28]	M	675			25.0	28.4
0.18 C–5.91 Mn–1.5 Si–0.4 Al-0.6Cr [28]	M	690			33.0	32.6
0.18 C–5.91 Mn–1.5 Si–0.4 Al-0.6Cr [28]	M	710			37.0	35.9

## Data Availability

The raw/processed data required to reproduce these findings cannot be shared at this time as the data form part of an ongoing study.

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
