# Peer review of "Process Maps for Predicting Austenite Fraction (vol.%) in Medium-Mn Third-Generation Advanced High-Strength Steels"

_materials, 2024, doi:10.3390/ma17050993_

Round 1

Reviewer 1 Report

Comments and Suggestions for Authors

1.- It is not clear why ferrite and martensite should be considered to behave as a single phase, because they have very different structural and thermidynamic properties. Furthermore, in the kinematic processes shown in Figure 1, it is very different if one or another structure is considered,

2.- The phrase "Based on the experimental observations conducted on med-Mn steels" and assuming a micron as the initial size does not seem to be appropriate for modeling, especially in this type of steels. In fact, it is too fine, since it is usually at least more than 5 microns, with a lower limit perhaps 2 microns, so it is suitable for the phenomenon analyzed.

3. It is necessary to establish the convergence of the calculations from the XRD shown

4.-  It is not possible to analyze figure 3 (incomplete)

5.- The experimental validation shown in Table 1 is insufficient, the authors apparently looked for bibliographical references that were close to their own predictions, which is not acceptable.

6.- The analysis of lines 376-386 does not seem very coherent with the results observed in the literature.

Reviewer 2 Report

Comments and Suggestions for Authors

The paper present an interesting application of Dictrra  to determination of retained austenited amount after intercrititical treatment of AHSS. I have the following comments:

In my opinion, it is necessary to explain more details of Dictra method. For instance, geometry, planar, spheroid, cylinder., number of nodes, phase composition, etc.

1)  It is necessary to specify more details of the Dictra simulation for instance, geometry of the analysis region, node numbers and boundary conditions.

2) The calculation was conducted using a constant austenite grain size of 1 micron, it seems too small and the grain size changes with temperature, Could you comment about this?

3) The Ms temperature can also be calculated with Thermo-Calc software, why did you use an empirical equation?

Reviewer 3 Report

Comments and Suggestions for Authors

The article presents the modeling of the structure of a prototype steel containing Mn depending on the annealing parameters. The presented research results are of a utilitarian nature for predicting the structure and properties of steel, especially for the automotive industry. These research results may be particularly useful in processes such as continuous galvanizing (mentioned by the authors), where high-temperature annealing is intended to prepare the sheet metal surface before applying the coating, and not to change the structure of the coating. However, this annealing, especially in a reduction furnace, can change this structure. For such cases, these modeling studies are extremely important.

The authors in this article did not conduct experimental studies to verify and validate the modeling results. However, they very accurately and accurately compared the obtained research results with the available data in References, which they also share. It's a pity that they didn't present their structure research in this article, but this is not a defect, just my suggestion.

There was also a good discussion in the article, which significantly increases the quality of the article.

Small Note:

Lines 128-132: “….which is compatible with industrial continuous galvanizing practice…”. The authors should explain why this is compatible with galvanizing practice because it is widely known that the continuous galvanizing process is carried out at a temperature of 460oC and the immersion time in the bath is several seconds. However, the processes to which the sheet metal is subjected before immersion in liquid zinc are not necessarily known. I think it would be good to supplement this information and explain, for example, why this modeling is important for the hot dip process. It is worth mentioning this in the introduction.

Round 2

Reviewer 1 Report

Comments and Suggestions for Authors

The manuscript can be accepted